# The Current State of Proteomics and Metabolomics for Inner Ear Health and Disease

**DOI:** 10.3390/proteomes12020017

**Published:** 2024-06-04

**Authors:** Motahare Khorrami, Christopher Pastras, Paul A. Haynes, Mehdi Mirzaei, Mohsen Asadnia

**Affiliations:** 1Faculty of Science and Engineering, School of Engineering, Macquarie University, Sydney 2109, NSW, Australia; motahare.khorrami@mq.edu.au (M.K.); christopher.pastras@mq.edu.au (C.P.); 2School of Natural Sciences, Macquarie University, Macquarie Park, Sydney 2109, NSW, Australia; paul.haynes@mq.edu.au; 3Department of Clinical Medicine, Faculty of Medicine, Health and Human Sciences, Macquarie Medical School, Macquarie University, Macquarie Park, North Ryde, Sydney 2109, NSW, Australia; mehdi.mirzaei@mq.edu.au

**Keywords:** inner ear, proteomics, metabolomics, mass spectrometry, meniere’s disease, ototoxicity, noise-induced hearing loss, vestibular schwannoma

## Abstract

Characterising inner ear disorders represents a significant challenge due to a lack of reliable experimental procedures and identified biomarkers. It is also difficult to access the complex microenvironments of the inner ear and investigate specific pathological indicators through conventional techniques. Omics technologies have the potential to play a vital role in revolutionising the diagnosis of ear disorders by providing a comprehensive understanding of biological systems at various molecular levels. These approaches reveal valuable information about biomolecular signatures within the cochlear tissue or fluids such as the perilymphatic and endolymphatic fluid. Proteomics identifies changes in protein abundance, while metabolomics explores metabolic products and pathways, aiding the characterisation and early diagnosis of diseases. Although there are different methods for identifying and quantifying biomolecules, mass spectrometry, as part of proteomics and metabolomics analysis, could be utilised as an effective instrument for understanding different inner ear disorders. This study aims to review the literature on the application of proteomic and metabolomic approaches by specifically focusing on Meniere’s disease, ototoxicity, noise-induced hearing loss, and vestibular schwannoma. Determining potential protein and metabolite biomarkers may be helpful for the diagnosis and treatment of inner ear problems.

## 1. Introduction

The mammalian inner ear is a remarkably sensitive signal detection system. This includes the mammalian cochlea, which can detect a wide bandwidth of sounds from 20 Hz to 20,000 Hz, spanning an extensive dynamic range of 140 dB (or 100,000,000,000 W of power) and including auditory cues from a whisper (20 µPa) to a jet engine’s roar (200,000,000 µPa) [1]. The inner ear can be divided into two parts: the membranous labyrinth and the bone labyrinth. (Figure 1). The membranous labyrinth is an intricate structure of fluid-filled tubes and chambers. It is found within the bone labyrinth and is separated into the vestibular and cochlear labyrinths [2]. The endolymphatic fluid fills the membranous labyrinth’s lumen with potassium, while the perilymphatic fluid, a sodium-enriched fluid with characteristics like those of extracellular fluid, fills the gaps around it [3]. The inner ear also includes balance organs, technically known as the vestibular system, which evolved before the hearing apparatus and encodes sensory stimuli relative to motions, such as angular rotations and linear accelerations, such as gravity [4,5]. Hearing and balance function is governed by mechanosensation—mechanical sounds and vibrations in the environment, which are the converted in the ear into all-or-none electrical signals, like binary code, which are sent to the brain for higher-order processing and perception or response. Mechanosensation relies on a well-controlled homeostatic environment involving hair cells, afferent neurons, supporting cells, and inner ear fluids containing potassium and sodium [6]. When the microenvironment of the inner ear changes, such as due to pathology, dysfunction of the sensory cells follows, resulting in debilitating hearing impairment and vestibular system disorders such as dizziness and vertigo.

Unfortunately, the cause of many inner ear disorders remains enigmatic [7]. One plausible explanation for this challenge is the lack of robust and accurate experimental approaches in animal models to rigorously study the pathophysiology of inner ear disease [8]. Further, direct recordings from inner ear hair cells and neurons are complex, as they are hidden in the skull behind the thick temporal bone, and exposing the end-organs through surgery changes their functional sensitivity, which is impossible in relatively healthy hearing subjects [9,10]. Another reason for the lack of progress in understanding inner ear pathology is the lack of biomarkers linked to specific diseases [11]. Consider cancer or Alzheimer’s disease, for example—both of these debilitating diseases have precise biomarkers and distinct biological boundaries defining the onset, progression, and severity of the disease, such as prostate-specific antigen and beta-amyloid (and/or tau) plaques (Alzheimer’s disease), and undergo comprehensive screening via imaging [12,13,14]. This is not the case with many inner ear disorders; the lines between inner ear dysfunction, diagnosis, and progression are blurry. A good example is Meniere’s disease, which is currently an idiopathic syndrome thought to involve multiple aetiologies, resulting in attacks of rotational vertigo lasting 20 min to 1 day, to progressive deafness, tinnitus, and aural fullness or pressure. It is not yet clear what the pathophysiology of Meniere’s disease is. Still, it is probably involved in the histopathological characteristics of the disease, called endolymphatic hydrops (ELH), which means the over-accumulation of endolymphatic fluid. The endolymphatic sac is the primary structure in the inner ear that forms and regulates endolymphatic fluid volume. A recent study indicated that SOX9 and SOX10 control the endolymphatic equilibrium and ensure the appropriate development of the endolymphatic system and ionic balance [15]. However, it is not yet known what the link between ELH and MD symptoms is, and progress has remained stagnant for decades, especially given the lack of reliable animal models with symptoms mimicking human MD. Unfortunately, there are no well-targeted treatments or cures due to a poor understanding of the aetiology and pathophysiology. For progress in early diagnosis, therapeutics, and cures, it is essential to isolate biomarkers or relevant pathological conditions resulting in inner ear disease and symptoms [16].

Several proteins and metabolites are abundant within the ear that perform distinct functions in forming and preserving various inner ear cells. Some of these biological molecules have been discovered to be feasible biological diagnosis markers for inner ear diseases. A challenge preventing the sampling of blood samples as a proxy for inner ear dysfunction is the blood–labyrinth barrier, which prevents proteins from entering the circulation from the inner ear [17]. To research the pathophysiology of inner ear illnesses, direct analysis of the perilymphatic and endolymphatic biofluid samples is the most effective technique [13]. One highly promising area to achieve this is via omics approaches, defined as the complete evaluation of genes (transcriptomics), proteins (proteomics), or metabolites (metabolomics) [18,19,20]. Investigations into the mammalian inner ear are now possible thanks to advances in proteomics- and metabolomics-based technologies that allow the identification and evaluation of hundreds to thousands of biomolecules in just a few microliters of bodily fluids [21].

Proteins, peptides, metabolites, lipids, glycans, and other biomolecules can all be studied using mass spectrometry (MS) techniques [22]. MS-based procedures can be categorised as ‘top-down’ and ‘bottom-up’, based on whether the proteins are analysed as whole molecules or after they have been digested into peptides. Although the association between the sequence of the proteins and its alterations may be preserved using ‘top-down’ proteomics techniques, ‘bottom-up’ approaches are among the most often used methods for the study of biological data because of their ability to be applied to both qualitative and quantitative workflows [23].

This review investigates the current state of research involving proteomics and metabolomics relevant to inner ear operation and pathology. We aim to provide a comprehensive review of the literature on the application of proteomic and metabolomic methods to understand inner ear disorders. Thus, our selection of articles was based on searching PubMed, Google Scholar and Science Direct using specific combinations of keywords, including proteomics, metabolomics, mass spectrometry, Meniere’s disease, ototoxicity, noise-induced hearing loss, and vestibular schwannoma. This approach enabled us to survey a large cross-section of the relevant recent literature. In addition, we highlight potentially promising protein and metabolite biomarkers, which may help to determine the cause of certain inner ear disorders.

## 2. Investigating Inner Ear Diseases through Proteomics and Metabolomics

According to the World Health Organization (WHO), 700 million people will have some degree of hearing loss by 2050 [24], while some reports have suggested that much higher levels of hearing loss may occur [25,26]. Regarding inner ear diagnosis and monitoring, providing a direct sample or biopsy is difficult due to the potential damage or destruction to the sensory organs. Therefore, limited information is available regarding the underlying mechanisms of inner ear damage [27]. Omics studies, which include genomics, transcriptomics, proteomics, metabolomics, and epigenomics, can reveal data at different molecular levels, which have significantly impacted our knowledge of biological phenomena. MicroRNAs (miRNAs) are short ribonucleic acids that control various biological functions in sensory areas, including the ears [28]. Identifying miRNAs in the human perilymphatic fluid could lead to discovering biomarkers for inner ear diseases. For instance, individuals suffering from Meniere’s disease display distinctive patterns of miRNA expression within the serum and perilymphatic fluid, potentially connected to postulated mechanisms for MD. It has been confirmed that miRNA-1299 levels are reduced in the serum and perilymphatic fluid of MD patients [29,30]. Proteomics and metabolomics could be valuable research approaches for investigating the biological changes in ear diseases, such as Meniere’s disease, ototoxicity, noise-induced hearing loss, and vestibular schwannoma. 

### 2.1. Meniere’s Disease (MD)

Meniere’s disease (MD) is an idiopathic inner ear disease distinguished by episodes of vertigo, unilateral fluctuating or permanent hearing loss, and tinnitus [31]. Acute attacks grow less frequent as the illness develops, but hearing loss and tinnitus become permanent, transitioning into a ‘burnt out’ phase of the disease with severe inner ear dysfunction. In addition, MD is known to be connected to endolymphatic hydrops, an accumulation of endolymphatic fluid in the endolymphatic space of the labyrinth [32]. MD is still a challenging medical condition to diagnose, especially in the beginning stages when not all common symptoms are present. As a result, determining the incidence of this illness is difficult [33]. However, MD’s prevalence has been predicted to range from 17 to 513 people per 100,000, which may be due to population differences in sampling and diagnostic criteria [34]. In terms of diagnosis, it is reported that functional indicators including pure tone audiometry (PTA; hearing threshold monitoring across frequency), electrocochleography (ECochG; direct electrical measurements from cochlear nerves), vestibular evoked myogenic potential (VEMP; vestibular driven reflex potentials from the eyes and neck), and magnetic resonance imaging (MRI) of hydrops within the inner ear, might be helpful in the evaluation of MD patients with various progressive disease stages [35].

In this context, proteomic analysis of biofluid samples provides an excellent opportunity for researchers to study the pathophysiology of MD [36]. Several studies have investigated protein profile changes in MD patient samples. In 2012, Chiarella et al. obtained 12 plasma samples from MD patients and eight plasma samples from healthy individuals. They compared protein expression in MD patients versus the control group by utilising 2D gel electrophoresis and subsequent LC-MS/MS analysis. It was found that the expression of complement factors H and B, fibrinogen alpha and gamma chains, beta-actin, and pigment epithelium-derived factors was elevated, while the expression levels of beta-2 glycoprotein-1, vitamin D binding protein, and apolipoprotein-1 were substantially decreased in the MD patient plasma [37].

Evidence suggests that inflammatory reactions that may occur in the inner ear and the immune system may be associated with the pathophysiology of MD [38]. Kim et al. sampled endolymphatic sac luminal fluid from three patients diagnosed with MD. LC-MS/MS was applied to analyse proteins isolated from 1-DE gels to assess the protein composition of MD patients’ fluid compared to controls. As a result, 180 proteins were identified and quantified in this work. Nine differentially abundant proteins were identified: eight consisted of immunoglobulin and its variants, and one was interferon regulatory factor 7 which was known to be involved in MD pathology [39]. In another study utilising LC-MS, Lin et al. conducted a comparative proteomic study of normal and MD patients’ perilymphatic fluid, four previously unknown protein biomarker candidates were discovered in the adult inner ear, namely AACT, HGFAC, EFEMP1, and TGFBI [40]. In 2021, Schmitt et al. presented the first comparative perilymphatic fluid proteome analysis in patients with three different inner ear diseases: an enlarged vestibular aqueduct, otosclerosis, and Meniere’s disease. Proteomics based on LC-MS and data-dependent acquisition was used to examine samples obtained during cochlear implantations. These studies highlighted that the protein short-chain dehydrogenase/reductase family 9C member 7 (SDR9C7) was only found in MD patients’ perilymphatic fluid and thus might be considered a potential biomarker [41]. A recent study focused on significant differences in the protein expression of peripheral blood mononuclear cells (PBMCs) in sporadic MD patients. The result indicated that endocytosis might have an impact on the pathophysiology of sporadic MD, and CHMP1A, VPS4A, FCN3, and MMP9 could be regarded as possible biomarkers [42].

Investigation of the inner ear metabolome may enable researchers to gain a greater understanding of its involvement in auditory system function as well as discover biomarkers that may be used to predict the responses to therapeutic interventions. In 2018, the human perilymphatic fluid’s metabolome was assessed by LC-MS, and 22 metabolites were detected, namely asparagine, lactic acid, valine carnitine, trigonelline, creatinine, glutamine, alanine, hypoxanthine, phenylalanine, sorbic acid, suberic acid, alpha-D-glucose, proline, 5-hyroxylysine, histidine, O-acetyl-l-carnitine, adipic acid, 3-methyglutaric acid, pimelic acid, N-acetyl-l-leucine, and arginine. Among them, asparagine and lactate had the strongest signals [43]. Moreover, Huang et al. performed a metabolomics study of ES luminal fluid (ELF) from patients with MD. By applying LC-MS in comparison with controls followed by statistical analysis, they showed that the MD patient samples contained elevated levels of hyaluronic acid, 4-hydroxynonenal, 2,3-diaminopropanoate, (5-L-glutamyl)-L-amino acid, D-ribulose 1,5-bisphosphate, 3-hydroxy-5-phosphonooxypentane-2,4-dione, and L-capreomycidine, along with a reduced level of citrate, ethylenediaminetetraacetic acid (EDTA), inosine 5′-tetraphosphate, D-octopine N-acetyl-D-glucosamine (GlcNAc), D-glucuronic acid (GlcUA), L-arginine, and 1-hydroxy-2-methyl-2-butenyl 4-diphosphate. These findings suggest that the increase in 4-hydroxynonenal could be associated with oxidative damage and inflammatory lesions in MD. Additionally, the decrease in citrate and EDTA might be linked to disturbed endolymphatic Ca^2+^ flux in MD patients [44].

### 2.2. Ototoxicity

Ototoxicity is a pharmaceutical side effect that causes cochlear or vestibular dysfunction of the inner ear receptor hair cells. Drug-induced inner ear dysfunction can manifest itself in a variety of ways: tinnitus, hearing loss, hyperacusis, auditory fullness, dizziness, and vertigo [45]. The most typical ototoxic pharmaceuticals used in the clinic include aminoglycoside antibiotics, platinum-based chemotherapeutic treatments (cisplatin and carboplatin), loop diuretics, macrolide antibiotics, and antimalarials [46]. Ototoxicity affects every age group; however, cisplatin ototoxicity has been shown to occur in between 23% and 50% of adults and in up to 60% of children. Ototoxicity usually develops undetected until a significant hearing difficulty with communication emerges. It is clinically detected by matching audiometric test results obtained before and after the ototoxic drug’s delivery [45].

However, proteomics studies in animal models recognise several potential protein biomarkers. For this purpose, in vivo proteome profiling was conducted on Sprague Dawley rats to analyse the effect of cisplatin on the cochlea. Cisplatin, as a platinum-containing inorganic compound, is an anti-cancer drug used in the treatment of several solid tumours [47]. Jamesdaniel et al. applied an antibody microarray to recognise early protein expression profiles in the cochlear tissue of control and cisplatin-treated rats. Physiological and morphological analysis showed an early stage of pathogenesis in animal models. In addition, they found differential expressions of 15 novel proteins that were engaged in apoptosis, cell survival, or cell cycle growth. The increased abundance of ATF2, JAB1, Mdm2, Rsk1, SUMO-1, myosin VI, p21WAF1Cip1, PRMT4, and reelin and the decreased abundance of active caspase 3 were correlated with a survival response. Moreover, overexpression of four proteins, including Tal, granzyme B, SLIPR/MAGI3, and RIP, and the decrease in the expression of epidermal growth factor (EGF), p35, and ubiquitin C-terminal hydrolase L1 were related to cell death reactions [48]. A later study by Waissbluth et al. was conducted on Sprague Dawley female rats to analyse the effect of cisplatin on the cochlea. Ten female rats in each cisplatin and control group received intraperitoneal (IP) cisplatin and saline, respectively, for three days. Then, the cochlea was extracted, and proteome analysis using SDS page fractionation and LC-MS was performed. As a result, the most remarkable proteome change was shown in Rab2A and Rab6A expression, with 33- and 25-fold decreases in expression [49]. In addition, research has been conducted to analyse changes in the metabolite profile caused by cisplatin. Laurell and colleagues treated guinea pigs with a cisplatin ototoxic dosage and then analysed the serum via LC-MS. Four distinct metabolite alterations were discovered to be strongly associated with high-frequency loss of hearing, including N-acetylneuraminic acid, L-acetylcarnitine, ceramides, and cysteinylserine [50]. Another study used cisplatin, administered intraperitoneally to rats; GC-MS and LC-MS were applied to discover potential plasma metabolite biomarkers. According to a GC/MS study, cysteine-cystine and 3-hydroxybutyrate, and based on LC/MS results, acylcarnitine (AC) 14:0, AC 18:1, AC 18:2, and phosphatidylethanolamine (PE) 18:2-18:2, were found to be differentially metabolome candidates. These findings facilitate the early diagnosis of cisplatin-induced nephrotoxicity [47].

Recently, Cui et al. evaluated the metabolic alterations in mice following vancomycin treatment. After vancomycin treatment, they used the chromatography-mass spectrometry (GC-MS) method to identify metabolites in serum and different organs, including the inner ear. Regarding the serum sample, urea, uracil, serine, myo-inositol, L-threonine, L-proline, L-phenylalanine, L-isoleucine, L-alanine, and glycine were upregulated. At the same time, 4-hydroxyproline, malic acid and L-valine were downregulated following the treatment. In terms of the inner ear, the only downregulated protein was undecane, whereas the expression of 6 proteins, including arachidonic acid, docosahexaenoic acid, gamma-butyrolactone, myo-inositol, ribitol and urea, were increased [51].

### 2.3. Noise-Induced Hearing Loss (NIHL)

Noise-induced hearing loss is a kind of sensorineural hearing impairment that starts at higher frequencies (3000–6000 Hz) and progresses slowly due to the involvement of loud sounds over a long period [52]. In many countries, this is the second most typical cause of acquired hearing loss after presbycusis. It is also estimated that around 1.1 billion young individuals have noise-exposed hearing impairment [53]. Despite the complexity of noise-induced hearing impairment pathogenesis, some efforts have been made to identify promising biomarkers.

In 2011, two different studies were conducted on the protein composition of NIHL animal models. In the first study, the cochlear tissue of four noise-induced chinchillas was removed separately. After employing an antibody microarray to understand the early changes in proteins, the results indicated an increase in the expression of FAK p–Tyr577, E2F3, hMps1, serine-threonine protein phosphatase 1b, activated p38/MAPK, WSTF, and Fas in the sensory epithelium. In contrast, the expression of E2F3, tropomyosin, CD146, and hnRNPA1 decreased in the lateral wall. In the modiolus, a significant increase was observed in the expression of aurora B, BID, HDAC10, and ADAM17, while a decrease occurred in cytokeratin 8 12, PRMT1, serine threonine protein phosphatase 2 A/B, NG2, brain nitric oxide synthase, DEDAF, and plakoglobin. Noise exposure triggered apoptosis in the sensory epithelium and modiolus, but not in the lateral wall [54]. In the second study, Yeo et al. compared protein expression profiles in control and noise-exposed BALB/C male mice. The control group was set up in a sound-free booth, whereas the experimental group underwent exposure to noise at a frequency of 1024/min for three hours on three separate days. The results indicated that the expression of seven proteins increased only in the cochlea of noise-exposed mice, including angiopoietin-like 1, heat shock 70 kDa protein, tyrosine-protein kinase MEG2, NaDC-1, myeloid Elf-1-like factor, ALCAM, metalloproteinase domain 7, and disintegrin [55]. Ten years later, Miao et al. used tandem mass tag (TMT)–labelling proteomics with LC-MS/MS to investigate protein expression alterations in the mouse model of NIHL. The differentially expressed proteins (DEPs) in the cochleae of noise-exposed and healthy mice were shown to be engaged in innate immune response pathways, the organisation of the extracellular matrix, and the viral defence response. After noise exposure, the results showed that pro-inflammatory cytokines such as TNF-α and IL-6 were considerably higher in mouse cochleae. Furthermore, various inflammation autophagy-related DEPs (ITGA1, KNG1, CFI, FGF1, AKT2, and ATG5) have been linked with NIHL [56]. In another study, the human cochlear proteins accompanied by NIHL pathophysiology were assessed by means of proteomic analysis. Serum samples from mining-based industrial workers were studied through one- or two-dimensional electrophoresis along with LC-MS/MS and MALDI-TOF-MS. The result provided fourteen proteins with abundance changes correlated with NIHL, including transthyretin, E3 ubiquitin protein ligase, ALB protein (albumin-like protein), transferrin, kininogen 1, enkurin and serpin peptidase inhibitor clade A (alpha 1 antiproteinase antitrypsin) member 3 isoform CRA_b, Plexin domain-containing protein 1, DNA oxidative demethylase, trifunctional purine biosynthetic protein adenosine 3, protein UNC 45, lysine specific demethylase 3A, coiled-coil domain-containing protein 62, and Myo 15 (Myosin) [57]. These candidate proteins might be employed as early diagnosis biomarkers and may help in our understanding of the NIHL progress.

In terms of metabolomic studies, a comparative investigation between guinea pigs’ inner ear fluid and plasma revealed that metabolite levels changed in the inner ear fluid. In contrast, the metabolite level in plasma was not affected after sound exposure. By applying chromatography/mass spectrometry (GC/MS) to the inner ear fluid, it was shown that the levels of ten metabolites including 3-hydroxy-butyrate, glycerol, fumaric acid, galactosamine, pyruvat + oxalacetic acid, phosphate, meso-erythritol, citric and isocitric acid, mannose, and inositol were increased after exposure to noise for 2 h, relative to the control group [58]. In a subsequent study, untargeted metabolomics was performed on noise-exposed guinea pig perilymphatic fluid. Among 15 discovered metabolites, pantothenic acid, creatine, butyrylcarnitine, acetylcarnitine, and two unidentified acylcarnitines, U137 and U569, had a higher intensity in the noise-exposed group [59].

In 2019, Ji et al. applied liquid chromatography-coupled tandem mass spectrometry (LC-MS/MS)-based metabolomics to assess metabolite abundance changes in the noise-exposed mouse. After analysing the inner ear tissue, 220 metabolites were identified that were correlated with glycolysis, mitochondrial metabolism, amino acids, and nucleotide metabolic pathways in the central carbon metabolism. However, 40 metabolites were differentially impacted by noise, among which 25 metabolites were upregulated and 15 metabolites were downregulated. The upregulated metabolites were nucleotides, cofactors, and carbohydrates, as well as alanine, aspartate, purine, glutamine, and glutamate metabolism, while the downregulated metabolites were involved in amino acid metabolism such as methionine, arginine, phenylalanine, tyrosine, and tryptophan [60]. Recently, another study on a mouse animal model suggested that 17 differential metabolites were considerably affected after 4 h of noise exposure. According to GC/MS metabolomics assessment, three metabolites, namely spermidine, 3-hydroxybutyric acid, and orotic acid, were significantly increased in the noise-contacted cochlea group in comparison with the reference group [61].

Boullaud et al. conducted metabolomic analysis through liquid chromatography coupled with high-resolution mass spectrophotometry to investigate early metabolic alterations in sheep perilymphatic fluid following noise-induced hearing loss. Twelve perilymphatic fluid samples were obtained: six from ears with normal hearing and six from NIHL ears exposed to noise. This model made it possible to identify 213 metabolites, including metabolites upregulated following noise exposure, such as urocanate, oleate, 5-oxo-L-proline, N-acetyl-glucose, N-acetylneuraminate, L-tyrosine, trigonelline, leukotriene-B4, 5,6-dihydrouracil, and 3-ureidopropionate. In addition, some components downregulated after noise exposure included deoxycarnitine, L-carnitine, N-acetyl-L-leucine, S-(5′-adenosyl)-L-homocysteine, and epinephrine. These results suggested that metabolic pathways such as oxidative stress, neuronal injury, and mechanical degradation were linked to noise exposure damage and hearing impairment [62].

Zhang et al. analysed a plasma sample of 60 noise-exposed workers through HPLC-MS/MS regarding human metabolomic profiles. Based on the comparative results between noise-exposed and control groups, Pro-Trp, adenine, dimethylglycine, 7 alpha-hydroxy dehydroepiandrosterone, calciferol, cis-5-dodecenoic acid, and 7 alpha-dihydroxy-5-cholestenoic acid were introduced as potential diagnostic markers. In addition, these results suggest that noise exposure could result in the dysregulation of different biological pathways, including immune response (differentiation of Th1 and Th2 cells, phagocytosis mediated through Fc-gamma receptors (FcγRs), cell cytotoxicity by natural killers) and the cell death procedure (apoptosis and necroptosis) [63]. Another human study highlighted that the autophagy pathway is essential to auditory cell fate and may be associated with NIHL progression and development. In this study, plasma samples of occupationally noise-exposed workers were collected after 12 h of fasting, and then LC-MS with electrospray ionisation in both the positive and negative ion modes were used for metabolomic profiling. As a result, the expression levels of organic acids such as homodeoxycholic acid and quinolacetic acid, along with 3,4-dihydroxymandelic acid, were found to be elevated, whereas those of phosphatidylethanolamine (15:0/20:2(11Z,14Z)), phosphatidylcholine (15:0/18:1(11Z)) and phosphatidylinositol (O-20:0/18:0) were reduced. Pathway analysis also showed that glycerophospholipid metabolism, glycosylphosphatidylinositol-anchor biosynthesis, choline metabolism, alpha-linolenic acid metabolism, linoleic acid metabolism, the retrograde endocannabinoid pathway, and autophagy were linked to NIHL [64].

Recent research conducted on noise-induced male rats also indicated metabolomic alteration after noise exposure. Assessment of serum samples through LC-MS showed that 1-oleoyl-2-palmitoyl-sn-glycero-3-phosphocholine, 3-hydroxybutyric acid, Pi 38:4, and Pe 38:4 were increased. In contrast, indolelactic acid, hippuric acid, 2,6-dihydroxybenzoic acid, 7-keto-3a, 12-a-dihydroxycholanic acid, acetaminophen sulfate, isatin, and quillaic acid were decreased in the noise-exposed group. Furthermore, seven pathways including glycosaminoglycan biosynthesis—keratan sulfate, N-glycan biosynthesis, the adipocytokine signalling pathway, peroxisome, bile secretion, various types of N-glycan biosynthesis, and the cAMP signalling pathway were shown to be dysregulated after noise exposure [65].

### 2.4. Vestibular Schwannoma (VS)

Vestibular schwannoma (VS) is the most prominent cerebellopontine angle (an anatomically complex region of the brain) tumour in adults that originates from the vestibular nerves [66,67,68]. In more than 90% of patients, the most typical signs include ipsilateral sensorineural hearing loss, dizziness, or imbalance in up to 61% of patients and asymmetric tinnitus in 55% of patients. Epidemiological studies have shown that the total incidence of VS is 1.4 per 100,000 people per year and has remained generally steady [69].

In 2011, Lysaght et al. used LC-MS/MS to compare the protein composition of human perilymphatic fluid collected from patients with vestibular schwannoma (VS) and patients undergoing cochlear implantation (CI). Two proteins, including μ-crystallin (CRYM) and low density lipoprotein-related protein 2 (LRP2), were detected in VS samples but not in the CI sample, which were considered as potential biomarkers in VS patients [70].

Cerebrospinal fluid (CSF) proteomic investigations have identified biomarkers for several intracranial cancers. In a pilot study, CSF samples were taken from individuals having surgery for untreated sporadic. The CSF proteins identified were similar to previously described VS biomarkers from secretions of tumours and the perilymphatic fluid, as well as a previously published normal CSF proteome. Chitinase 3–like protein was a promising biomarker in VS samples, with elevated abundance in half of the samples analysed. There was only one protein, fibronectin 1, which was differentially expressed in VS and overlapped with the putative proteins in the perilymphatic fluid and tumour secretions [71]. Another study explored potential VS progression markers based on the CSF proteome of 43 patients with various VS grades. After utilising isobaric tags for relative and absolute quantitation (iTRAQ)-based and LC-MS/MS proteomics techniques and following that with ELISA assays, the results indicated SCG1, KLF11, and CA2D1 as early biomarkers of VS. ABCA3 and KLF11 upregulation as well as BASP1 and PRDX2 downregulation in CSF were also correlated with VS growth in the early phase [72]. Later investigation on the perilymphatic fluid’s protein composition of 15 VS patients revealed that alpha-2-HS-glycoprotein was detectable in all patients as an independent variable for tumour-related hearing loss [73]. In a study conducted by Xu et al., iTRAQ labelling and LC-MS/MS were applied to find the candidate proteins that were differentially expressed among VS and healthy human vestibular tissues that were possibly associated with cell proliferation and apoptosis. Six candidates were significantly upregulated, including LGALS1, ANXA1, ANXA2, GRB2, STAT1, and SPARC, while one protein, CAV1, was the most significantly downregulated. Western blotting and immune histochemistry assays confirmed that the expression of LGALS1, ANXA1, GRB2, and STAT1 was increased dramatically in VS [74]. In addition, Seo et al. investigated the sporadic forms of vestibular schwannoma (VS) pathophysiology through protein analysis of two vestibular nerves and two VS human tissues. In this study, two-dimensional electrophoresis was used, followed by MALDI-TOF MS analysis. The findings indicated 29 proteins with different expressions, seven of which were associated with apoptosis, including upregulated Annexin V, Annexin A4, Annexin A2 isoform 2, YWHAZ protein, ARHGDIA, HSP27, and downregulated peroxiredoxin 6. The results were confirmed by means of Western blotting and immunohistochemistry analysis, which showed that apoptosis is potentially correlated with the VS pathophysiology [75]. Based on the neuroradiological appearance, VS is classified into two subtypes, namely solid VS (SVS) and cystic VS (CVS), which are distinguished by aggressive clinical signs such as fast tumour growth [76]. In 2020, Tandem Mass tag (TMT) isotopic labelling quantitative proteomics analysis was performed to discover the proteins that change in expression between CVS and SVS. COL1A1 and COL1A2 proteins were recognised as potential indicators for diagnosing and treating CVS. This suggested that reduced collagen deposition could be one of the reasons why CVS patients are vulnerable to bleeding [77].

Researchers have not yet studied the alteration of metabolomes in SV patients, although it has been demonstrated that the analysis of metabolomic changes is associated with several types of tumours and can be used for diagnosis or therapeutic approaches [78,79,80,81].

Table 1 and Table 2 provide a summary of the currently identified proteomic and metabolomic biomarkers in the abovementioned inner ear disorders.

Although there were no shared markers between different studies for each disease, the proteomic analysis revealed that the fibrinogen alpha chain protein was upregulated in both MD and ototoxicity diseases. In terms of metabolomic studies, the upregulation of O-acetyl-l-carnitine and trigonelline and the downregulation of citrate, L-arginine and D-glucuronic acid have been shown between MD and NIHL. Furthermore, the upregulation of 3-hydroxy-butyrate was reported in both ototoxicity and NIHL.

It is worth noting that omics technology provides insights into the biological mechanisms of other significant ear disorders, such as age-related hearing loss (ARHL) and sudden sensorineural hearing loss (SSNHL).

Presbycusis, also known as ARHL, is the most frequent age-related otologic problem that mainly impacts mobility, with a prevalence of more than 75% in those aged 70 and older [82]. The combination of neurosensory hearing loss (loss of hair cells or neurons) and conductive hearing loss (middle ear abnormalities) could cause ARHL [26]. In a recent study following successful ARHL modelling using C57BL/6, mice from the Old (48-week-old) and Youth (4-week-old) groups had their cochleae subjected to quasi-targeted metabolomics compared to analysis of the metabolic changes and associated pathways. As a result, ergothioneine, S-adenosyl-L-methioninamine, and histamine were shown to be upregulated, whereas DL-citrulline, isocytosine and L-citrulline were significantly downregulated [83].

In addition, SSNHL is characterised by a sudden loss of hearing of ≥30 dB in three continuous frequencies within 72 h. The SSNHL incidence varies between 5 to 20 per 100,000 individuals. Wang et al. analysed the serum metabolic profile of 20 SSNHL patients and 20 healthy controls using LC-MS. Among the top ten distinctive metabolites between the two groups, guanosine, 3-hydroxycapric acid, tumonoic acid A, Prolyl-tyrosine, and Nylidrin were downregulated, while Ribothymidine, octanoylcarnitine, moxisylyte, chaetoglobosin N, and sphingosine were upregulated [84].

## 3. Recent Progress and Future Directions in Inner Ear Omics

MS enables researchers to perform investigations into the proteome and distinct post-translational modifications (PTMs) [85]. Proteomics techniques are significantly impacted by proteoforms, which are unique protein molecules produced by a single gene because of changes in transcription, translation, and PTMs [86]. PTMs in proteoforms play essential roles in cell signalling, protein degradation, and other biological procedures. Mass spectrometry is the major tool for examining proteoforms’ PTMs. The complexity of proteomes, which are composed of proteoforms, poses challenges to bottom-up proteomics (BUP), while top-down proteomics (TDP) can discover intact proteoforms [87]. To overcome the limitations of BUP, proteomics researchers have focused on developing tools for the direct analysis of intact proteoforms using TDP, which can cover a wide range regarding the masses and physicochemical properties of proteoforms [88].

The capacity to directly examine the developmental processes of complex tissues and complete organisms has been made possible by single-cell MS-based proteomics [89]. Zhu et al. investigated progressive alterations in the proteins found in developing hair cells from vestibular gravity organs (the utricle) of embryonic day 15 chickens. The single-cell proteomic analysis results indicated that the expression levels of proteins such as OCM, CRABP1, GPX2, AK1, and GSTO1 were elevated, whereas TMSB4X and AGR3 were decreased during hair cell development. In addition, single-cell transcriptome analysis revealed similar mRNA alterations during hair cell development. Therefore, single-cell proteomics findings may help to demonstrate cellular and developmental characteristics that transcriptomics may have missed [90].

In addition, a recent study conducted by Arambula et al. investigated previously published information to determine if individuals with MD, otosclerosis, an enlarged vestibular aqueduct, sudden hearing loss, and hearing impairment with an unknown cause (controls) have distinct perilymphatic fluid proteome profiles. Analysing perilymphatic fluid proteins obtained from 81 patients suffering from various ear disorders with the benefit of using a machine learning method demonstrates the enrichment of MD perilymphatic fluid proteins in a variety of internal cochlear cells, such as spiral ganglion neurons and stria vascularis cells and some external cells that are a part of the immune response pathway [91]. The main challenge in this field is identifying the best datasets involving large sample numbers and diverse sample types. After collecting biofluids with the appropriate criteria, machine learning could help to find meaningful markers. However, the application of machine learning in the field of inner ear disorders is a relatively new area of research, and information on the appropriate sample size for training artificial intelligence (AI) can be limited.

However, there are several challenges in the field of proteomics-based biomarker discovery, which refer to the gap between discovery and validation. Several factors that contribute to the inability to validate findings from the discovery phase have been pointed out. These factors include issues with the discovery sample set, such as inadequate size, absence of suitable controls, and variations in the patient population between the discovery and validation experiments [92,93]. Another difficulty might be distinguishing between inter-individual variation and experimental noise in protein profiling. Even with these challenges, global proteome profiling could serve as a valuable instrument for identifying biomarkers if incorporated into a straightforward analysis approach [94,95].

The cochlea’s fragility makes direct sampling challenging. An indirect biopsy using inner fluids is needed to discover innovative therapies for inner ear illness. This could improve cochlear implants and assess progressive hearing loss and specialised treatments [36,96]. In terms of various samples for studying the inner ear environment, it is evident that blood represents a significant part of the inner ear, and sustained microcirculation is essential for the normal function of the inner ear. Like the blood–brain barrier, the inner ear contains an endothelial blood–tissue barrier that highly controls ion transport in and out [97]. The blood–labyrinth barrier represents a biological barrier between the vasculature and inner ear fluids which regulates protein and water flow between the blood and inner ear fluids. Systemic blood is a possible source for inner ear sampling because, under some situations, chemicals in the blood can pass through the BLB and enter the inner ear [98,99]. Blood tests represent a simple and non-invasive approach to identifying the most specific proteins in the inner ear structures and differential metabolite changes caused by damage. It has been shown that inner ear-related biomarkers are detected in the circulating blood, plasma, or serum when they pass across the blood–brain barrier, including otolin-1, prestin, and matrilin-1 [11,100].

However, one of the significant limitations of detecting biomarkers of localised disorders, like inner ear disorders, is that the biomarkers must be obtained from the fluid in direct contact with the injured organ. Metabolites in the blood or CSF might come from other tissues in the body; thus, metabolomic screening of inner ear fluid may be more suitable than whole-blood samples. Although many proteins are shared between the perilymphatic fluid, endolymphatic fluid, and plasma, several proteins are specific to inner ear fluids and CSF in significantly high concentrations. As a result, these proteins offer massive potential for research into normal and pathological inner ear systems. However, it would be very difficult to have non-invasive access to the inner ear fluid [13,101]. Animal models are beneficial, mainly if different species are employed and online bioinformatics techniques are used to identify potential human homologues [36].

In conclusion, there is little comparative research on the various samples that can be used to determine which is most promising for diagnosis application. Defining the best sample in terms of preparation procedure, specificity, and sensitivity to help diagnose inner ear diseases might be the way to go in future studies. Advancements in sample preparation methods and mass spectrometry instrumentation might provide a platform for studying inner ear tissues or challenging-to-acquire body fluids.

## Figures and Tables

**Figure 1 proteomes-12-00017-f001:**
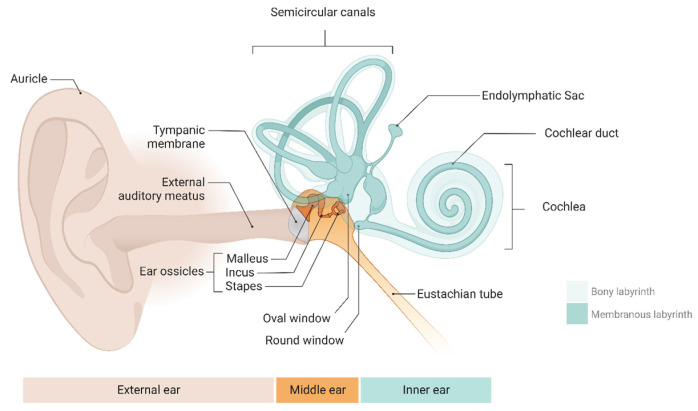
The anatomy of the human ear includes the external, middle, and inner ear (comprising the cochlea and vestibular system). Created with BioRender.com.

**Table 1 proteomes-12-00017-t001:** Summary of proteomics research related to inner ear disorders, including Meniere’s disease, ototoxicity, noise-induced hearing loss, and vestibular schwannoma.

Disease	Participants	Sample	Method	Expression Protein Changes/Key Proteins	Ref
Meniere’s disease (MD)	Human	Plasma	2-DE and LC-MS/MS	Upregulation of complement factors H and B, fibrinogen alpha and gamma chains, beta-actin, and pigment epithelium-derived factors Downregulation of beta-2 glycoprotein-1, vitamin D binding protein, and apolipoprotein-1	[37]
		Endolymphatic fluid	1-DE and LC-MS/MS	Upregulation of immunoglobulin (Including IgM, Ig kappa light chain variable region, Ig heavy chain variable region; VH3 family, Ig heavy chain VHDJ region, AF1 non-allergic IgE heavy chain IGHV3-74) and interferon regulatory factor 7	[39]
		Perilymphatic fluid	SDS-PAGE and HPLC-MS/MS	Upregulation of AACT, HGFAC, EFEMP1, and TGFBI	[40]
			LC-MS	Upregulation of short-chain dehydrogenase/reductase family 9C member 7 (SDR9C7)	[41]
Ototoxicity	Rat	Cochlear tissue	Antibody microarray	Upregulation of ATF2, JAB1, Mdm2, Rsk1, SUMO-1, myosin VI, p21WAF1Cip1, PRMT4, reelin, Tal, granzyme B, SLIPR/MAGI3 and RIP Downregulation of active caspase 3, EGF-epidermal growth factor, p35, and ubiquitin C-terminal hydrolase L1	[48]
			SDS-PAGE and LC-MS/MS	Upregulation of Ba1-647, fibrinogen alpha chain isoform 2 precursor, tropomyosin-1, perlecan (heparan sulfate proteoglycan 2), Ab2-131, acid ceramidase precursor and Alpha-parvin. Downregulation of Rab2A, Rab6A, Cd81, ribosomal protein S5, isoform CRA_b, myelin basic protein, glycerol-3-phosphate dehydrogenase [NAD+], Ras-related protein Rap-1b precursor, H2A histone family (member X and member Y2), tenascin-R precursor, and eosinophil peroxidase precursor,	[49]
Noise-induced hearing loss (NIHL)	Chinchillas	Cochlear tissue	Antibody microarray	Upregulation of FAK p–Tyr577, E2F3, hMps1, serine-threonine protein phosphatase 1b, activated p38/MAPK, WSTF and Fas, aurora B, BID, HDAC10, and ADAM17 Downregulation of E2F3, tropomyosin, CD146, hnRNPA1, cytokeratin 8 12, PRMT1, serine-threonine protein phosphatase 2 A/B, NG2, brain nitric oxide synthase, DEDAF and plakoglobin	[54]
	Mouse	Cochlear tissue	2-DE and MALDI-TOF MS	Upregulation of angiopoietin-like 1, heat shock 70 kDa protein, tyrosine-protein kinase MEG2, NaDC-1, myeloid Elf-1-like factor, ALCAM, metalloproteinase domain 7, and disintegrin.	[55]
		Cochlear tissue	TMT-labelling and LC-MS/MS	Upregulation of TNF -α, IL-6, ITGA1, KNG1, CFI, Downregulation of FGF1, AKT2, and ATG5	[56]
	Human	Serum	2-DE and LC-MS/MS	Plexin domain-containing protein 1, DNA oxidative demethylase, trifunctional purine biosynthetic protein adenosine 3, protein UNC 45, lysine specific demethylase 3A, coiled-coil domain-containing protein 62, and Myo 15 (Myosin)	[57]
			2-DE and MALDI-TOF MS	Upregulation of transthyretin, E3 ubiquitin protein ligase, albumin, transferrin, kininogen 1, enkurin, and serpin peptidase inhibitor clade A (alpha 1 antiproteinase antitrypsin) member 3 isoform CRA_b	[57]
Vestibular Schwannoma (VS)	Human	CSF	iTRAQ and MS/MS	Upregulation of fibronectin 1, chitinase 3–like protein 1 precursor, clusterin preproprotein, gelsolin isoform a precursor, Ig lambda-like polypeptide 5 isoform 1, haemoglobin subunit alpha, and haptoglobin isoform 1 preproprotein Downregulation of alpha-1-acid glycoprotein 1 precursor, Lysozyme C precursor, Secretogranin-1 precursor, and Keratin	[71]
			iQTRAQ and LC-MS/MS	Upregulation of apolipoprotein A–I (APOA1), ABCA3, CA2D1 and KLF11 Downregulation of BASP1 and PRDX2	[72]
		Perilymphatic fluid	LC-MS	Expression of alpha-2-HS-glycoprotein	[73]
		Vestibular tissue	iTRAQ and LC-MS/MS	Upregulation of LGALS1, ANXA1, ANXA2, GRB2, STAT1, and SPARC Downregulation of CAV1	[74]
		Vestibular tissue	2-DE and MALDI-TOF MS	Upregulation of Annexin V, Annexin A4, Annexin A2 isoform 2, YWHAZ protein, ARHGDIA, and HSP27 Downregulation of Peroxiredoxin 6	[75]
		Tissue	TMT-labelling and LC-MS/MS	Downregulation of COL1A1 and COL1A2 in CVS patients	[77]

LC-MS/MS, liquid chromatography-tandem mass spectrometry; 2-DE, two-dimensional gel electrophoresis; 1-DE, one-dimensional gel electrophoresis; SDS-PAGE, sodium dodecyl-sulfate polyacrylamide gel electrophoresis; MALDI-TOF-MS, matrix-assisted laser desorption/ionisation-time of flight mass spectrometry; TMT, tandem mass tag; iTRAQ, isobaric tags for relative and absolute quantitation.

**Table 2 proteomes-12-00017-t002:** Summary of metabolomics research related to inner ear disorders, including Meniere’s disease, ototoxicity, and noise-induced hearing loss according to the publication journal, participants, sample, techniques, and differential expression profile of metabolites.

Disease	Participants	Sample	Method	Key Findings	Ref
Meniere’s disease (MD)	Human	Perilymphatic fluid	LC-MS	Upregulation of asparagine, lactic acid, valine carnitine, trigonelline, creatinine, glutamine, alanine, hypoxanthine, phenylalanine, sorbic acid, suberic acid, alpha-D-glucose, proline, 5-hydroxylysine, histidine, O-acetyl-l-carnitine, adipic acid, 3-methyglutaric acid, pimelic acid, N-acetyl-l-leucine, and arginine	[43]
		Endolymphatic sac luminal Fluid	LC-MS/MS	Upregulation of hyaluronic acid, 4-hydroxynonenal, 2,3-diaminopropanoate, (5-L-glutamyl)-L-amino acid, D-ribulose 1,5-bisphosphate, 3-hydroxy-5-phosphonooxypentane-2,4-dione, and L-capreomycidine Downregulation of citrate, EDTA, inosine 5′-tetraphosphate, D-octopine N-acetyl-D-glucosamine (GlcNAc), D-glucuronic acid (GlcUA), L-arginine, and 1-hydroxy-2-methyl-2-butenyl 4-diphosphate	[44]
Ototoxicity	Guinea pig	Serum	LC-MS	Upregulation of N acetylneuraminic acid, L-acetyl carnitine, ceramides, and cysteinyl serine	[50]
	Rat	Plasma	LC/MS	Upregulation of 3 acylcarnitine and a phosphatidylethanolamine with C18:2–C18:2	[47]
			GC/MS	Upregulation of cysteine–cystine and 3-hydroxy-butyrate	[47]
Noise-induced hearing loss (NIHL)	Guinea pig	Inner ear fluid	GC/MS	Upregulation of 3-hydroxy-butyrate, glycerol, fumaric acid, galactosamine, pyruvat + oxalacetic acid, phosphate, meso-erythritol, citric acid, isocitric acid, mannose, and inositol	[58]
		Perilymphatic fluid	HILIC-UHPLC-Q-TOF–MS	Upregulation of pantothenic acid, creatine, butyryl carnitine, acetylcarnitine, two unidentified acylcarnitine, U137, and U569	[59]
	Mouse	Cochlea and vestibular organ tissue	LC-MS/MS	Upregulation of pyridoxal 5-phosphate, inosine 5-monophosphate, inosine-5 phosphate, uridine-monophosphate, cytidine monophosphate, sucrose, L-aspartate, xanthosine 5-monophosphate, guanosine 5-monophosphate, adenosine 5-monophosphate, O-acetyl-l-carnitine, D-fructose 6-phosphate, oxidised glutathione, N-methyl-l-glutamate, NAD, aminoadipate, adenosine 3,5-diphosphate, adenosine 5-diphosphate, cytidine 5-diphosphocholine, flavin adenine dinucleotide, L-glutamic acid, succinate, N-acetyl-l-aspartic acid, cytosine, and adenosine Downregulation of uracil, L-leucine, L-phenylalanine, L-ornithine, D-ornithine, D-glucuronic acid, citrulline, L-tryptophan, L-arginine, xanthurenic acid, L-methionine, DL-isocitric acid, citrate, and adenosine-diphosphoglucose	[60]
		Cochlear tissue	GC/MS	Upregulation of spermidine, 3- hydroxybutyric acid, and orotic acid	[61]
	Sheep	Perilymphatic fluid	LC/MS	Upregulation of urocanate, Oleate, 5-oxo-L-proline, N-acetyl-glucose, N-acetylneuraminate, L-tyrosine, trigonelline, leukotriene-B4, 5,6-dihydrouracil, and 3-ureidopropionate Downregulation of deoxycarnitine, L-carnitine, N-acetyl-L-leucine, S-(5′-Adenosyl)-L-homocysteine, and epinephrine.	[62]
	Human	Plasma	HPLC-MS/MS	Upregulation of 7 alpha-hydroxy dehydroepiandrosterone Downregulation of pro-Trp, adenine, dimethylglycine, calciferol, cis-5-dodecenoic acid, and 3 beta, 7 alpha-dihydroxy-5-cholestenoic acid	[63]
			UPLC/Q-TOF-MS	Upregulation of homodeoxycholic acid, quinolacetic acid, and 3,4-dihydroxy mandelic acid Downregulation of phosphatidylethanolamine, phosphatidylcholine, and phosphatidylinositol	[64]
	Rat	Serum	UPLC/Q-TOF-MS	Upregulation of 1-oleoyl-2-palmitoyl-sn-glycero-3-phosphocholine, 3-hydroxybutyric acid, Pi 38:4 and Pe 38:4 Downregulation of indolelactic acid, hippuric acid, 2,6-dihydroxybenzoic acid, 7-keto-3-alpha, 12-alpha-dihydroxycholanic acid, acetaminophen sulfate, isatin, and quillaic acid	[65]

LC-MS/MS, liquid chromatography-tandem mass spectrometry; LC-MS, liquid chromatography- mass spectrometry; GC-MS, gas chromatography-mass spectrometry; HILIC-UHPLC-Q-TOF–MS, hydrophilic interaction chromatography-ultra-high-pressure liquid chromatography mass spectrometry; HPLC-MS/MS, high performance liquid chromatography-tandem mass spectrometry; UPLC-Q-ToF-MS, ultra-high performance liquid chromatography quadrupole time-of-flight mass spectrometry.

## Data Availability

No new data were created or analysed in this paper.

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
