# Peer review of "The Current State of Proteomics and Metabolomics for Inner Ear Health and Disease"

_proteomes, 2024, doi:10.3390/proteomes12020017_

Round 1

Reviewer 1 Report

Comments and Suggestions for Authors

The presentation presents four areas (Meniere's disease, ototoxicity, noise-induced hearing loss, and vestibular schwannoma) that are all investigate for diagnosis with biomarkers.  Overall, the presentation is well written and provides a broad perspective with omics and fluid analysis. Much of the analysis is going on but can be better by presenting certain more recent citations (see below).

P36 ..ear can be divided..

59 suggest at least one additional paper, for example: Petitpre et al., (2022)  13:3878

64 suggest citing one work: Rolls, Edmund T., et al.  Cerebral Cortex 33.10 (2023): 6207-6227.

71; A very good citation of hydrops would be a great presentation:  Szeto, Irene YY, et al. Proceedings of the National Academy of Sciences 119.46 (2022): e2122121119. Note that the Sox9/10 deletions provides a novel approach to the hydrops.  I suggest to reword the introduction once you have fully appreciate this paper.

90 Perhaps it would be a great presentation with a recent review on genetics of the auditory system: Sun…Z Liu (2022) Cell reports 38; 110542

92-154 nice and short presentation of flow diagram.

157 Most recent evidence suggest a much higher hearing loss: Jiam + Rauch (2023) Frontiers in Neuroscience, 17, 1169122’ Elliott, Karen L., et al. Frontiers in aging neuroscience 14 (2022): 814528.

294 cite here the work by Elliott et al., 2022, that details the loss of OHCs in the base..

406 a good citation would be the recently published data that details the role of schwannoma: Kersigo, Jennifer, et al. "Effects of Neurod1 expression on mouse and human schwannoma cells." The Laryngoscope 131.1 (2021): E259-E270.

462-557. I suggest reconsidering the presentation to about 1.5 papers.  I know the work has already outlined earlier and can reduce the presentation somewhat.

Author Response

Response to Reviewer 1 Comments

The authors express gratitude for the reviewers' efforts and time spent reviewing our manuscript. Given the useful suggestions and comments made by the reviewers, we have revised the original manuscript to address the issues identified in the review report. We hope this revision has substantially improved the manuscript to the satisfaction of the reviewers.

The authors’ responses to specific comments, suggestions, or queries are indicated below.

  1. P36 ..ear can be divided..

Response:

Changed as requested.

  1. 59 suggest at least one additional paper, for example: Petitpre et al., (2022)  13:3878

Response:

This citation was added on page 2, line 60. The text now reads:

… which is impossible in relatively healthy hearing subjects [9,10].

Where 10 is the reference by Petitpre.

  1. 64 suggest citing one work: Rolls, Edmund T., et al. Cerebral Cortex10 (2023): 6207-6227.

Response:

This citation was added at line 65 as requested. The text now reads:

…and comprehensive screening via imaging [12-14]

Where 14 is the reference by Rolls.

  1. A very good citation of hydrops would be a great presentation:  Szeto, Irene YY, et al. Proceedings of the National Academy of Sciences46 (2022): e2122121119. Note that the Sox9/10 deletions provides a novel approach to the hydrops.  I suggest to reword the introduction once you have fully appreciate this paper.

Response:

We have updated the endolymphatic hydrops section with the recommended paper on page 2, lines 71-76.

The text now reads:

Endolymphatic hydrops (ELH), which means the over-accumulation of endolymph. The endolymphatic sac is the primary structure in the inner ear that forms and regulates endolymph volume. A recent study indicated that SOX9 and SOX10 control endolymph equilibrium and ensure the appropriate development of the endolymphatic system and ionic balance [15].

  1. 90 Perhaps it would be a great presentation with a recent review on genetics of the auditory system: Sun…Z Liu (2022) Cell reports 38; 110542.

Response:

This citation was added on page 3, line 94.

The text now reads:

… proteins (proteomics), or metabolites (metabolomics) [18-20]

where 120 in the reference by Sun et al

  1. 92-154 nice and short presentation of flow diagram.

Response:

Thank you.

  1. 157 Most recent evidence suggest a much higher hearing loss: Jiam + Rauch (2023) Frontiers in Neuroscience, 17, 1169122’ Elliott, Karen L., et al. Frontiers in aging neuroscience14 (2022): 814528.

Response:

We have added this information and the relevant references on page 5lines167-168, so the text now reads:

According to the World Health Organization (WHO), 700 million people will have some degree of hearing loss by 2050 [40], while some reports have suggested much higher levels of hearing loss may occur [41,42].

  1. 294 cite here the work by Elliott et al., 2022, that details the loss of OHCs in the base..

Response:

We have added the reference and some explanation to the age-related hearing loss section on page 13, lines 499-502. The text now reads:

Presbycusis, also known as ARHL, is the most frequent age-related otologic prob-lem that mainly impacts mobility, with a prevalence of more than 75% in those 70 and older [104]. The combination of neurosensory hearing loss (loss of hair cells or neurons) and conductive hearing loss (middle ear abnormalities) could cause ARHL [42].

Where 42 is the reference by Elliott et al.

  1. 406 a good citation would be the recently published data that details the role of schwannoma: Kersigo, Jennifer, et al. "Effects of Neurod1 expression on mouse and human schwannoma cells." The Laryngoscope1 (2021): E259-E270.

Response:

This reference was added to the Vestibular schwannoma (VS) section on page 10, line 435. The text now reads:

Vestibular Schwannoma (VS) is the most prominent cerebellopontine angle [an anatomically complex region of the brain] tumor in adults that originates from the vestibular nerves [88-90].

Where 90 is the reference by Kersigo et al.

  1. 462-557. I suggest reconsidering the presentation to about 1.5 papers.  I know the work has already outlined earlier and can reduce the presentation somewhat.

Response:

Thanks for the suggestion. However, this section has already been truncated, and we believe its current length appropriately conveys the relevant information regarding recent work and future directions of inner ear omics.

Reviewer 2 Report

Comments and Suggestions for Authors

This is a review of current state of knowledge about attempts to identify molecular biomarkers of inner ear doseases. The authors aimed to broadely review the field of proteome and metabolome reserach in Menier's disease (MD), ototoxicity, noise-inducrs hearing loss (NIHL), vestibular schwannoma (VS). They reviewed both animal and human studies, looking for both peripheral biomarkers (plasma, serum) and organ biomarkers (perilymph, endolymph, cochlear tissue in animal cases), as well as relations between them. The topic is very interesting and promising in the diagnostics, though difficult, with most difficulties connected with access to the diseased, not to mention healthy inner ear.

I found it well organised and easy to follow rewiev, with vast citations showing broad expertise in the field of the methodology and results. However, there are several v good, contributing reports on research on human perilymph, which I cannot see included in the review:

1. Lysaght A.C, et al. The  proteome if Human perilymph. doi: 10.1021/pr200346q.

2.Shew M et al Feasibility of microRNA profiling in human inner ear perilymph. doi: 10.1097/WNR.0000000000001049.

3. Shew M et al. Distinct MicroRNA profiles in the perilymph and serum of patients with Meniere's Disease. doi: 10.3389/fneur.2021.646928.

and some further form these groups.

Reviewer 3 Report

Comments and Suggestions for Authors

Khorrami et al in the review untitled “The current state of proteomics and metabolomics for inner ear health and disease” summarized the knowledge about the role of potential diagnosis biomarkers in inner ear disorders through the development of omics methods as proteomics and metabolomics. The authors specifically focused four inner ear disorders: Meniere’s disease, ototoxicity, noise-induced hearing loss and vestibular schwannoma.

However, I have comments and suggestions to improve the review.

Major concerns:

The authors mentioned neither sensorineural sudden hearing loss (SSHL) nor age-related hearing loss (ARHL or presbycusis). It would be better to also have a focus on presbycusis that affects 1 in 3 older people. Presbycusis is a major public health problem. What about ARHL/SSHL and proteomics? What about ARHL/SSHL and metabolomics (for example, PMID:36148011)?

I think that Part2 can be removed. The Figure 2 with a detailed legend must be sufficient to summarize the general strategies of proteomics.

The authors did not describe the methods how they chose their articles. Did they use the PRISMA methodology (PMID: 33782057). A flow diagram summarizing the study selection process would be helpful.

Minor concerns:

Some references are missing; In Meniere’s disease part, the authors did not mention one reference about biomarkers (Lin et al., 2023: PMID:37603046). In ototoxicity part, the authors can add a study that suggests the role of Vancomycin (Cui et al., 2022; PMID:35855991).

The recent review written by Malesci et al. is absent in the manuscript (PMID:37894867).

Line 34: write 1011

Line 35: write 2*108

Reviewer 4 Report

Comments and Suggestions for Authors

Please revise the manuscript to elevate its impact.  

Round 2

Reviewer 1 Report

Comments and Suggestions for Authors

The paper has significantly improved to present a broad variety of Meniers and related vestibular and cochlear losses with age.  The paper is well thought out and has additional citations.  Since the latest addition to deal with miR's, I suggest adding one paper to detail with the  miRs

line 174: Suggesting a paper that details the role of specifc function in null mutations (see the work of Kersigo et la Genesis 49.4 (2011): 326-341.
